# Generation of Fibrotic Liver Organoids Using Hepatocytes, Primary Liver Sinusoidal Endothelial Cells, Hepatic Stellate Cells, and Macrophages

**DOI:** 10.3390/cells12212514

**Published:** 2023-10-24

**Authors:** Yongdae Yoon, Seong Chan Gong, Moon Young Kim, Soon Koo Baik, Ju-Eun Hong, Ki-Jong Rhee, Hoon Ryu, Young Woo Eom

**Affiliations:** 1Regeneration Medicine Research Center, Yonsei University Wonju College of Medicine, Wonju 26426, Republic of Korea; yongdae0611@naver.com (Y.Y.); drkimmy@yonsei.ac.kr (M.Y.K.); baiksk@yonsei.ac.kr (S.K.B.); 2Department of Surgery, Yonsei University Wonju College of Medicine, Wonju 26426, Republic of Korea; surgeon_g@yonsei.ac.kr; 3Department of Internal Medicine, Yonsei University Wonju College of Medicine, Wonju 26426, Republic of Korea; 4Department of Biomedical Laboratory Science, College of Software and Digital Healthcare Convergence, Yonsei University Mirae Campus, Wonju 26493, Republic of Korea; jehong@yonsei.ac.kr (J.-E.H.); kjrhee@yonsei.ac.kr (K.-J.R.)

**Keywords:** liver organoid, fibrosis, inflammation, Matrigel, three-dimensional culture

## Abstract

Liver organoids generated with single or multiple cell types have been used to investigate liver fibrosis development, toxicity, pathogenesis, and drug screening. However, organoid generation is limited by the availability of cells isolated from primary tissues or differentiated from various stem cells. To ensure cell availability for organoid formation, we investigated whether liver organoids could be generated with cell-line-based Huh-7 hepatocellular carcinoma cells, macrophages differentiated from THP-1 monocytes, and LX-2 hepatic stellate cells (HSCs) and primary liver sinusoidal endothelial cells (LSECs). In liver organoids, hepatocyte-, LSEC-, macrophage-, and HSC-related gene expression increased relative to that in two-dimensional (2D)-cultured Huh-7/LSEC/THP-1/LX-2 cells without Matrigel. Thioacetamide (TAA) increased α-smooth muscle actin expression in liver organoids but not in 2D-cultured cells, whereas in TAA-treated organoids, the expression of hepatic and LSEC markers decreased and that of macrophage and HSC markers increased. TAA-induced fibrosis was suppressed by treatment with N-acetyl-L-cysteine or tumor-necrosis-factor-stimulated gene 6 protein. The results showed that liver toxicants could induce fibrotic and inflammatory responses in liver organoids comprising Huh-7/LSEC/macrophages/LX-2 cells, resulting in fibrotic liver organoids. We propose that cell-line-based organoids can be used for disease modeling and drug screening to improve liver fibrosis treatment.

## 1. Introduction

Liver fibrosis is a pathological condition caused by extracellular matrix (ECM) accumulation driven by alcohol abuse, viral infections, and hepatotoxic drugs. The end stage of liver fibrosis is cirrhosis, for which liver transplantation is the gold standard treatment [1]. However, owing to various limitations, including a shortage of donor livers, stem cell therapy has been considered an alternative treatment to alleviate liver fibrosis [2,3]. As mesenchymal stem cells (MSCs) modulate immune cell activities through the expression of various factors such as nitric oxide, prostaglandin E2, indoleamine 2,3-dioxygenase, interleukin (IL)-6, IL-10, and HLA-G; induce hepatic progenitor or stem cell proliferation; and transdifferentiate into hepatocytes, clinical studies on treating liver fibrosis using MSCs are actively being conducted [4,5,6]. However, the exact in vivo mode of action of MSCs and their interactions with liver cells are unclear.

Hepatocytes, the most abundant cells in liver tissue, are involved in central functions, such as detoxifying exogenous substances, the urea cycle, protein synthesis, and lipid and carbohydrate metabolism in the liver [7]. During liver injury, damaged hepatocytes undergo apoptosis and release damage-associated patterns (DAMPs), activating macrophages and hepatic stellate cells (HSCs) and recruiting lymphocytes [8,9]. DAMPs induce cytokine secretion in Kupffer cells and liver sinusoidal endothelial cells (LSECs), resulting in HSC activation and LSEC dysfunction [10,11]. In addition, tumor necrosis factor (TNF)-α released by dying Kupffer cells activates HSCs and LSECs, transiently producing chemokines and adhesion molecules that orchestrate monocyte engraftment. Engrafted circulating monocytes transmigrate into the perisinusoidal space and come into close contact with hepatocytes. Therefore, cell–cell interactions between hepatocytes, HSCs, LSECs, and Kupffer cells, as well as their secreted factors, are essential in the fibrotic progression in the liver [12,13].

Organoids are self-organized three-dimensional (3D) cell cultures with distinct organotypic features [14,15]. Liver organoids have been derived from single-type cell cultures, usually hepatocytes, or multi-type cell co-cultures, such as those using hepatocytes, HSCs, endothelial cells, Kupffer cells, and mesenchymal cells [16]. Organoids can be used for liver disease modeling or to generate functional hepatocytes to repopulate damaged livers. Liver organoids using EPCAM+ biliary epithelial cells generate functional hepatocytes with hepatobiliary features that could be directed toward a hepatocyte-like phenotype [17,18]. Recently, human and mouse hepatocytes were cultured long term as organoids and successfully engrafted in damaged mouse livers [19]. Liver organoids comprising multiple cell types show structure formation, joint function, inflammatory responses, and improved cellular phenotypes compared with those of monoculture cells [20,21,22]. Liver buds with hepatic endoderm cells differentiated from induced pluripotent stem cells, MSCs, and human umbilical vein endothelial cells (HUVECs) can be vascularized, resulting in functional human liver cells after ectopic transplantation into immunodeficient mice [23]. Three-dimensional multicellular microtissue co-cultures with human cell lines representing hepatocytes (HepaRG), THP-1 macrophages, and stellate cells have been designed for liver fibrosis modeling. Hepatocellular injury, antioxidant defense responses, and activation of Kupffer cells and HSCs lead to ECM deposition in 3D-multicellular microtissues treated with pro-fibrotic compounds, such as transforming growth factor-beta 1, methotrexate (MTX), and thioacetamide (TAA) [24]. In addition, Leite et al. developed liver organoids using HepaRG and primary human HSCs, observing hepatocyte injury-dependent HSC activation after MTX, allyl alcohol, or acetaminophen (APAP) treatment [25].

Many studies that developed liver organoids have used stem-cell-derived hepatocytes, HSCs, and mesenchymal cells. However, these studies were limited by low hepatocyte functions, spontaneous HSC activation, and high inter-batch variability [21,22,26,27]. In addition, cell–cell communication between hepatocytes, HSCs, LSECs, and Kupffer cells is critical for liver fibrosis progression [12,13]. However, research has not been conducted on creating liver organoids with these four cells. In this study, cell-line-based hepatocytes (Huh-7), THP-1-derived macrophages, stellate cells (LX-2), and primary endothelial cells (LSECs) were used to prepare liver organoids. We further investigated whether fibrosis could be induced efficiently and if the activity of these four cell types could be altered. In addition, to determine whether it was possible to develop an evaluation system for MSCs through an accurate understanding of therapeutic mechanisms, we verified whether cell-line-based liver organoids could be used for drug screening.

## 2. Materials and Methods

### 2.1. Cell Culture

Huh-7 and THP-1 cells were purchased from the Korea Cell Line Bank (Seoul, Republic of Korea) and maintained in Roswell Park Memorial Institute Medium 1640 (RPMI 1640, Gibco BRL, Rockville, MD, USA) with 10% fetal bovine serum (FBS; Gibco BRL) and penicillin/streptomycin (Gibco BRL). LX-2, an HSC line, was obtained from Millipore (Burlington, MA, USA) and subcultured in Dulbecco’s Modified Eagle Medium (DMEM; Gibco BRL) with 3% FBS and penicillin/streptomycin. LSECs and HUVECs were purchased from ScienCell (Carlsbad, CA, USA) and cultured with an endothelial cell medium (ScienCell). LSECs and HUVECs at passage four were used in this study because in LSECs beyond five passages, α-SMA expression increases and CD31 expression decreases. Immortalized primary hepatocytes and LSECs were purchased from BioIVT (Westbury, NY, USA) and maintained in Upcyte^®^ Hepatocyte High-Performance Medium and Upcyte^®^ LSEC Culture Medium (BioIVT), respectively. All cells were incubated at 37 °C and 5% CO_2_.

### 2.2. Liver Organoid Generation

A mixture of RPMI 1640 and endothelial cell medium in equal proportions was used to generate liver organoids. Matrigel (Corning Life Sciences, Corning, NY, USA) was thawed overnight at 4 °C and mixed with a cold medium in a 1:1 ratio on ice. Diluted Matrigel (50 µL per well) was added to a 96-well plate and placed in a cell culture incubator for 10 min. Macrophages were differentiated from THP-1 monocytes with phorbol 12-myristate-13-acetate (100 nM) for two days. Thereafter, Huh-7 (4 × 10^4^ cells; or BioIVT’s immortalized primary hepatocytes), LSECs (4 × 10^4^ cells; BioIVT’s immortalized primary LSECs or HUVECs), macrophages (1 × 10^4^ cells), and LX-2 (1 × 10^4^ cells) were prepared in warm medium (150 µL) and then seeded on a Matrigel-coated 96-well plate. We referenced the optimal cell ratios from Takebe et al. to generate cell-line-based liver organoids [23]. Subsequently, the medium was replaced with fresh medium every two days. To determine whether TGF-β plays an important role during liver organoid generation, TGF-β receptor inhibitor A83-01 was added at a concentration of 500 nM on day 0. For 2D cultures, Huh-7, LSECs, macrophages, and LX-2 cells were cultured in a 96-well plate without a Matrigel coating. To induce fibrosis, liver organoids were treated with 1 mM of TAA, 100 mM of Et-OH, 10 mM of APAP, or 1 ng/mL of TGF-β on day 3 for an additional three days. To determine changes in fibrosis or inflammatory response, TAA- or TGF-β-induced fibrotic liver organoids were treated with 1 mM of N-acetyl-L-cysteine (NAC) or 40 ng/mL of TNF-stimulated gene 6 protein (TSG-6) on day 4.

### 2.3. Immunoblotting

Liver organoids were lysed using sample buffer (62.5 mM Tris-HCl, pH 6.8, 34.7 mM sodium dodecyl sulfate (SDS), 5% β-mercaptoethanol, and 10% glycerol), sonicated, boiled for 5 min, subjected to SDS-polyacrylamide gel electrophoresis, and transferred onto a polyvinylidene difluoride membrane (Millipore). Sonication was performed using a VCX 500 sonicator (Sonics, Newtown, CT, USA) with 20% amplitude for 4 s in a 1 s on/1 s off mode. The membrane was blocked with 5% skim milk in Tris-HCl-buffered saline containing 0.05% Tween 20 (TBST) for 30 min and incubated with primary antibodies against α-SMA (ab7817 or ab5694; Abcam, Cambridge, UK), COL1A1 (#39952; Cell Signaling Technology, Danvers, MA, USA), CD31 (sc376764; Santa Cruz Biotechnology, Santa Cruz, CA, USA), and GAPDH (sc47724; Santa Cruz Biotechnology) overnight at 4 °C. The membrane was washed thrice for 5 min with TBST and incubated with horseradish peroxidase-conjugated secondary antibodies (7074S and 7076S; Cell signaling Technology) for 2 h. After washing thrice with TBST, the bands were treated with an EZ-Western Lumi Pico or Femto (Dogen, Seoul, Republic of Korea) and detected using a ChemiDoc XRS+ System with Image Lab^TM^ software Version 5.2.1 (Bio-Rad Laboratories, Hercules, CA, USA).

### 2.4. Quantitative Polymerase Chain Reaction (qPCR)

Total RNA was prepared from liver organoids treated with TRIzol reagent (Gibco BRL). cDNA was synthesized from total RNA (1 µg) using a Verso cDNA Synthesis Kit (ThermoFisher Scientific, Waltham, MA, USA). The cDNA was mixed with a primer and power SYBR Green PCR Master Mix (Applied Biosystems, Dublin, Ireland) and amplified using a QuantStudio 6 Flex Real-time PCR system (ThermoFisher Scientific). The primer sequences are indicated in Appendix A. The 2^−(△△Ct)^ method measured relative fold changes in mRNA expression.

### 2.5. Immunofluorescence Staining

Liver organoids were fixed in 4% paraformaldehyde (Tech and Innovation, Chuncheon, Korea) for 1 h and incubated in 30% sucrose phosphate buffer overnight at 4 °C. Thereafter, the liver organoids were embedded with FSC 22 frozen section medium (Leica Biosystems, Wetzlar, Germany) and cut into 10 µm-thick sections in a cryostat at −20 °C. Next, the sections were fixed with 4% paraformaldehyde for 10 min, permeabilized with 1% Triton X100 in phosphate-buffered saline (PBS) for 10 min, and blocked using blocking buffer (5% bovine serum albumin, 10% FBS, and 0.2% Triton X100 in PBS) for 30 min. Primary antibodies against CK8 (sc-8020; Santa Cruz Biotechnology), CD31 (GA610; Dako, Glostrup, Denmark), EMR1 (sc-377009; Santa Cruz Biotechnology), and PDGFRα/β (ab32570; Abcam) in blocking buffer were incubated with the sections overnight at 4 °C. For fluorescence labeling, the sections were incubated with Alexa Fluor 488-conjugated secondary antibody (1:100; Invitrogen, Carlsbad, CA, USA) for 1 h at 25 °C. Nuclear staining was conducted using DAPI (4′,6-diamidino-2-phenylindole; Sigma-Aldrich, St. Louis, MO, USA), and samples were mounted using DPX Mounting Medium (Electron Microscopy Sciences, Hatfield, PA, USA). The sections were observed and imaged under a confocal microscope (LSM800; Zeiss, Oberkochen, German).

### 2.6. Picro-Sirius Red Staining

Liver organoids were fixed in 4% paraformaldehyde (Tech and Innovation) for 1 h at 4 °C and embedded in 3% agarose in phosphate buffer (pH 7.4; Tech and Innovation). The agarose-embedded samples were formed into paraffin blocks and cut into 4 µm-thick sections at Yonsei Immunohistopathology Lab (Wonju, Korea). Sections were deparaffinized in xylene twice for 3 min, 100% Et-OH twice for 2 min, 95% Et-OH for 1 min, and 70% Et-OH for 1 min, followed by tap water for 5 min. The sections were stained using a Picro-Sirius Red Stain Kit (Abcam) according to the manufacturer’s instructions and then mounted using DPX Mounting Medium (Electron Microscopy Sciences). The degree of fibrosis was observed under a light microscope, and images were obtained (Eclipse TS2R; Nikon, Tokyo, Japan).

### 2.7. Statistical Analyses

All statistical analyses were performed using one-way analysis of variance, followed by Tukey’s post hoc tests. A two-tailed Student’s *t*-test was used to evaluate the differences between two groups. Data are presented as the means ± standard deviation (SD). All *p* values < 0.05 were considered statistically significant.

## 3. Results

### 3.1. Liver Organoids Generated Using Multiple Cells

To generate liver organoids, Huh-7, LSECs, LX-2, and macrophages differentiated from THP-1 cells were seeded in Matrigel-coated 96-well plates (Figure 1A). Unlike cells in 2D cultures (Figure 1B), the four cell types cultured in Matrigel began to aggregate over time and showed a spherical shape on day 1. The spherical organoids became more densely aggregated on day 3 and maintained their shape until day 6 (Figure 1C). In 2D cultures, cell viability was significantly reduced after three days, but organoid viability did not change significantly until six days (Figure 1D). Furthermore, one cell type was excluded to determine whether the four cell types were necessary for self-organization when preparing liver organoids. Liver organoids did not form without LSECs or when LSECs were replaced with HUVECs (Figure 1E). Immortalized primary human hepatocytes and LSECs formed liver organoids with LX-2- and THP-1-differentiated macrophages (Figure 1F). To determine the presence of a distribution pattern of liver tissues in the generated organoids, immunofluorescence staining was performed with antibodies against hepatocytes (CK8), endothelial cells (CD31), macrophages (EMR1), and fibroblast (PDGFRα/β) markers. Notably, CK8-positive hepatocytes were observed in spherical clusters similar to cirrhotic nodules, while PDG-FRα/β-positive LX-2 cells were mostly located on the outer periphery of the organoids and in areas with fewer hepatocytes. CD31-positive LSECs were predominantly observed in hepatocyte-rich areas but, similar to LX-2 cells, were also observed on the periphery of organoids. EMR1-positive macrophages were mostly located near PDG-FRα/β-positive LX-2 cells (Figure 1G). These results suggest that LSECs are essential for self-organization, and Huh-7, LSECs, LX-2, and macrophage cells can self-organize with each other in Matrigel to form a structure similar to the nodule-like spheroid shape observed in cirrhosis.

### 3.2. Effects of TGF-β Signaling on Liver Organoid Generation

Among the four cell types, liver organoids were not formed without LSECs but were generated without LX-2 (Figure 1C). Nevertheless, we could not conclude whether LSECs were the most essential cells for liver organoid production as we used four times more LSECs than LX-2 cells for liver organoid generation. Takabe et al. showed that mesenchymal cells are important for organoid generation [23]. Therefore, we determined whether the self-assembly of organoids could be induced by mesenchymal-cell-activating TGF-β or by increasing the ratio of the mesenchymal LX-2 cells, even in the absence of LSECs. First, we evaluated whether TGF-β signaling is critical in organoid generation. Because LSECs in the liver secrete TGF-β and play an essential role in activating latent TGF-β, we used LSECs below passage four that did not show a decrease in CD31 or an increase in α-SMA expression (Figure 2A). During liver organoid generation, the TGF-β receptor inhibitor (A83-01) completely inhibited liver organoid formation using the four cell types (Figure 2B). These results suggest that the TGF-β signaling pathway can play a key role in liver organoid generation. Next, we investigated whether increasing the LX-2 cell ratio could induce liver organoid generation. When liver organoids were prepared with Huh-7/macrophages/LX-2 cells without LSECs, increasing the LX-2 cell ratio slightly increased self-assembly but failed to induce spherical organoid formation. However, TGF-β treatment increased the self-assembly of Huh-7/macrophages/LX-2 cells and enhanced spherical organoid formation, similar to liver organoids comprising Huh-7/LSECs/macrophages/LX-2 cells (Figure 2C). Taken together, our results suggest that both the TGF-β signaling pathway and mesenchymal cells are essential for liver organoid formation.

### 3.3. Gene Expression Profiles in 2D- and 3D-Cultured Cells

The expression of alpha-fetoprotein (*AFP*) increased in our liver organoids. In contrast, functional liver markers (such as albumin (*ALB*), hepatocyte nuclear factor 4 alpha (*HNF4α*), cytochrome P450 families (*CYPs*), glucose transporter 2 (*GLUT2*), the UDP glycosyltransferase 2 family (*UGT2B*), and glucose-6-phosphatase catalytic subunit 1 (*G6PC1*)) exhibited lower expression levels than those in the human liver. However, the levels of most functional liver markers increased in Huh-7/LSEC/macrophage/LX-2 cells cultured with Matrigel (3D-cultured cells or liver organoids) compared with those in 2D-cultured cells (Figure 3A). The levels of *Interleukin (IL)-1β*, *IL-6*, and *TGF-β* inflammation-related markers increased (Figure 3B), whereas those of α-smooth muscle actin (*α-SMA*) decreased. Most collagens (except *COL1A1* and *COL2A1*), Laminin-111 components (*LAMA1*, *LAMB1*, and *LAMC1*), fibronectin 1 (*FN1*), nidogen 1 (*NID1*), and elastin (*ELN*) showed increased expression in liver organoids (Figure 3C). The expression of lymphatic vessel endothelial hyaluronan receptor-1 (*LYVE1*), an LSEC-specific marker, increased, but that of *CD31*, a general endothelial cell marker, decreased (Figure 3D). Moreover, the expression of functional LSEC genes, such as vascular endothelial growth factor A (*VEGF-A*), vascular endothelial growth factor receptor 2 (*VEGFR2*), endothelial nitric oxide synthase (*eNOS*), and endothelin receptor type B (*EDNRB*), increased (Figure 3D). These results suggest that each cell type comprising the liver organoid cultured in Matrigel is more functional than 2D-cultured cells. In particular, the expression of α-SMA, a fibrosis marker, decreased despite a clear increase in the ECM, which is vital for the 3D structure of multiple cells or cell adhesion, suggesting that cell-line-based liver organoids may be similar to normal or early fibrotic liver cells.

### 3.4. Fibrosis Modeling in Liver Organoids Using Liver Toxicants

TAA is widely used to induce liver fibrosis in animal models. To investigate whether cell-line-based liver organoids respond to TAA-induced fibrosis, liver organoids at day 3 were treated with TAA for an additional three days. A concentration of 1 mM TAA or higher significantly increased α-SMA and COL1A1 expression in liver organoids (Figure 4A). However, α-SMA expression was not detected in liver organoids generated using Huh-7, LSECs, or macrophages, indicating that α-SMA expression was LX-2-dependent (Figure 4B). Furthermore, in 3D liver organoids, α-SMA expression increased dose-dependently at 1–10 mM TAA (Figure 4A,C). Nonetheless, it gradually decreased at >20 mM TAA, and no α-SMA expression was observed at 80 mM TAA (Figure 4C). α-SMA was not detected in 2D-cultured cells treated with TAA up to 80 mM (Figure 4C). Al-Bader et al. reported that optimal TAA concentrations induce liver cirrhosis in rats more effectively, while high TAA concentrations induce severe degenerative changes [28]. Therefore, we used 1 mM of TAA to induce fibrosis in liver organoids in this study.

As most collagens except *COL1A1* and *COL2A1* were increased in liver organoids (Figure 3C), Picro-Sirius Red staining was performed to detect collagens in liver organoids. TAA-treated organoids were more intensely and broadly stained than the control group (Figure 4D). Other liver toxicants, such as ethanol (Et-OH) and APAP, also increased α-SMA and COL1A1 expression in liver organoids (Figure 4E). These results suggest that cell-line-based liver organoids can become fibrotic due to hepatic toxicants, inducing α-SMA expression in an HSC-dependent manner.

### 3.5. Gene Expression Profiles Using Liver Toxicants

Although HSCs are the primary cells producing ECM in liver fibrosis, interactions between HSCs and surrounding cells are crucial for DAMP release, hepatic sinusoid capillarization, and inflammation during liver fibrosis [29]. Therefore, we investigated whether changes in major gene expression in the liver organoid constituent cells—Huh-7, LSECs, macrophages, and HSCs—were present after TAA treatment. In liver organoids treated with TAA, the levels of hepatocyte *AFP* and *ALB* markers decreased, but those of inflammatory (i.e., inducible nitric oxide synthase (*iNOS*), *IL-1β*, and *TNF-α*), fibrotic (i.e., *TGF-β*), and M2-macrophage markers (i.e., *CD206*) increased (Figure 5A,B). In addition, *α-SMA*, *COL1A1*, and *COL3A1* expression was significantly increased in TAA-treated liver organoids (Figure 5C). Furthermore, the levels of the endothelial cell marker *CD31* and vasoconstriction gene *EDNRB* increased, but those of functional endothelial cell markers, such as *LYVE1*, *VEGF-A*, *VEGFR2*, and *eNOS*, reduced (Figure 5D).

In addition, we investigated whether the levels of CYPs, which are enzymes for clearing hepatic toxicants, increased after TAA, APAP, or Et-OH treatment. Although the expression levels of the CYP superfamily differed according to the hepatic toxicants, the expression of most *CYPs* was significantly increased by TAA, APAP, and Et-OH (Figure 5E). The results further indicate that liver organoids treated with TAA induce inflammatory and detoxification responses and endothelial cell malfunction and produce ECM. These results indicate that our findings in cell-line-based liver organoids are similar to the pathology observed during liver fibrosis. Thus, when exposed to liver toxicants, the cells constituting liver organoids can comprehensively express the genes required for detoxification, inflammation, and fibrosis.

### 3.6. Validation of the Fibrotic Liver Organoid Model with N-Acetyl-L-Cysteine and Tumor-Necrosis-Factor-Stimulated Gene 6 Protein Treatment

As liver fibrosis is a wound-healing process accompanied by inflammation, oxidative stress, and fibrosis, therapeutic strategies have been designed to reduce oxidative stress using NAC or alleviate inflammation and fibrosis using TSG-6 [30,31]. Therefore, we investigated whether the therapeutic effects against fibrosis and inflammation could be verified in a fibrotic liver organoid model using NAC and TSG-6. First, fibrotic liver organoids induced by TAA were treated with NAC. As expected, the levels of α-SMA and COL1A1 proteins and those of *α-SMA*, *COL1A1*, and *COL3A1* mRNAs, which were increased by TAA in liver organoids, decreased to control levels by treatment with NAC (Figure 6A–C). In addition, NAC significantly decreased the expression of *iNOS*, *IL-1β*, *TGF-β*, *CD206*, *CYPs*, and *UGT2B7*, which are essential for regulating inflammation and detoxification in the liver (Figure 6D,E). These results show that TAA induces oxidative stress in liver organoids, which increases fibrosis, immune response, and liver detoxification; however, the reduction in oxidative stress with NAC can lessen the severity of liver damage. These results suggest that, similar to liver fibrotic animal models, cell-line-based liver organoids could be used to assess the therapeutic effect of NAC.

Next, we investigated whether TSG-6, which has anti-inflammatory and antifibrotic functions, could ameliorate fibrosis and inflammation in TAA-treated liver organoids. Similar to the therapeutic effects of NAC, TSG-6 reduced the protein and mRNA expression of *α-SMA*, *COL1A1*, and *COL3A1* in TAA-treated liver organoids (Figure 7A,B) and reduced the intensity of Picro-Sirius Red staining (Figure 7C). TSG-6 also suppressed inflammation-related gene expression (Figure 7D) and reduced *CYPs* and *UGT2B7* expression to control levels (Figure 7E). These results demonstrate the anti-inflammatory and antifibrotic effects of TSG-6 in TAA-treated liver organoids.

TGF-β increases ECM production in LX-2 cells through mothers against decapentaplegic homolog 3 (SMAD3) phosphorylation, and TSG-6 can regulate α-SMA expression by inhibiting SMAD3 phosphorylation [32]. Therefore, we analyzed whether TGF-β could induce fibrosis in liver organoids and whether TSG-6 could alleviate TGF-β-induced fibrosis. TGF-β-treated liver organoids exhibited increased expression of *α-SMA*, *COL1A1*, and *COL3A1* mRNA and α-SMA protein (Figure 8A,B), whereas TSG-6 decreased their expression (Figure 8C,D). Furthermore, TSG-6 reduced the SMAD3 phosphorylation increased by TGF-β in liver organoids (Figure 8E). In addition, Picro-Sirius Red staining confirmed that TSG-6 reduced TGF-β-dependent collagen expression (Figure 8F). These results suggest that TGF-β can induce fibrosis in cell-line-based liver organoids, and this model can be used to screen drugs that regulate the TGF-β signaling pathway.

TAA and TGF-β induced fibrosis in cell-line-based liver organoids, whereas NAC, which reduces oxidative stress, and TSG-6, which reduces inflammation and fibrosis, ameliorated inflammation and fibrosis in fibrotic liver organoids. These results suggest that cell-line-based liver organoids can mimic the pathology of liver fibrosis and can be used for drug screening to identify novel treatments for liver fibrosis.

## 4. Discussion

In this study, we observed that cell-line-based liver organoids can self-assemble in three days. Furthermore, treatment with TAA induced fibrosis in liver organoids in three days of treatment. Notably, we found that treating fibrotic liver organoids with NAC decreased inflammation and fibrosis. Similarly, TSG-6 treatment reduced inflammatory and fibrotic responses in the fibrotic liver organoids.

Huh-7, LSECs, macrophages, and LX-2 cells self-assembled in Matrigel to form liver organoids, but not when LSECs were absent. The expression of genes related to inflammation, fibrosis, detoxification, and LSEC function increased in liver organoids cultured in 3D compared with that in 2D-cultured cells. In addition, TAA amplified the expression of genes associated with these cellular processes when added to liver organoids, except for genes such as *AFP*, *ALB*, *LYVE1*, *VEGF-A*, *VEGFR2*, and *eNOS*. Picro-Sirius Red staining indicated that a substantial amount of collagen was expressed in liver organoids and increased in TAA-treated liver organoids. NAC and TSG-6 treatments alleviated TAA-induced fibrosis and inflammation in liver organoids. These results suggest that cell-line-based liver organoids can be applied to model liver diseases, including fibrosis, because they show responses such as inflammation, fibrosis, detoxification, and LSEC function regulation in response to liver toxicants. As NAC or TSG-6 alleviates inflammation and fibrosis in fibrotic liver organoids, cell-line-based liver organoids can be used to explore drugs for treating fibrotic diseases or to understand the mode of action of therapeutic agents, including MSCs. For example, liver organoid transplantation increased survival through differentiation into more functional liver cells in a liver failure mouse model. Liver organoids were generated using MSC-derived multiple liver cells, and these liver organoids expressed lower amounts of albumin (ALB) and CYP3A7 than those expressed by the adult liver but more than those by MSC-derived hepatocytes [33].

During liver disease progression, hepatic fibrosis and angiogenesis occur in parallel. Angiogenesis in mice after VEGF injection accelerates liver fibrosis, partly via the provision of latent TGF-β activated on the surface of HSCs by plasma kallikrein. Subsequently, the active TGF-β stimulates HSC activation [34,35,36]. Therefore, crosstalk between LSECs and HSCs can promote liver fibrosis onset and progression through VEGF and TGF-β. In our liver organoids, the expression of *VEGF-A*, *TGF-β*, and collagens, excluding *COL1A1* and *COL2A1*, was increased in 3D-cultured cells compared with that in 2D-cultured cells, but the expression of *α-SMA* was considerably decreased. In addition, collagen-rich areas stained with Picro-Sirius Red were observed in the control liver organoids. These results indicate that cell-line-based liver organoids may already be in the fibrosis initiation or progression stages.

An advantage of 3D culture is that a stable phenotype of each constituent cell can be maintained. When primary rat hepatocytes and rat LSECs were 3D-cultured, the expression of sinusoidal endothelial 1, a unique LSEC marker, was constantly maintained during the 12-day culture period, and hepatocytes showed a 3- to 6-fold increase in ALB and increased CYP1A1/2 and CYP3A activity [37]. In addition, when HepaRG cells were 3D-cultured with LSECs, the HepaRG cell viability and APAP conversion levels were the highest. However, when cultured with HUVECs, the HepaRG cell viability and ALB expression decreased, which could be due to the functions of hepatocytes that are influenced by the endothelial cell (EC) line in the 3D culture [38]. Moreover, LSECs, but not HUVECs, facilitated a stable ALB expression in primary human hepatocytes for 11 days, and when co-cultured with fibroblasts, each cell could maintain a stable phenotype for 3 weeks [39]. Consistent with these previous studies, in liver organoids comprising Huh-7, LSECs, macrophages, and LX-2 cells, increases in ALB, CYPs, and gene expression related to LSEC function were observed. In addition, compared with 2D-cultured cells, the level of α-SMA decreased, while the activity of most ECMs and inflammatory responses increased. Furthermore, our study demonstrated that liver organoid formation could be regulated according to the EC line, as liver organoids were not produced when HUVECs were used rather than LSECs. Although these results have limited application because the organoids were based on cell lines, cell-line-based liver organoids are expected to be used to quickly and accurately understand cell–cell communication and the pathophysiology of liver diseases. In the future, it will be necessary to compare and verify the utility of cell-line-based liver organoids through preparation using primary hepatocytes, HSCs, LSECs, and Kupffer cells and analyzing their gene expression and reactivity to toxic liver substances.

## 5. Conclusions

Based on our results, we conclude that liver organoids are rapidly generated using cell line-based Huh-7, LX-2, and THP-1 cells, and primary LSECs, and that fibrotic liver organoids are produced in response to liver toxicants. We expect that our fibrotic liver organoid model will be used for drug screening and to understand the exact therapeutic mechanisms of drugs, including MSCs.

## Figures and Tables

**Figure 1 cells-12-02514-f001:**
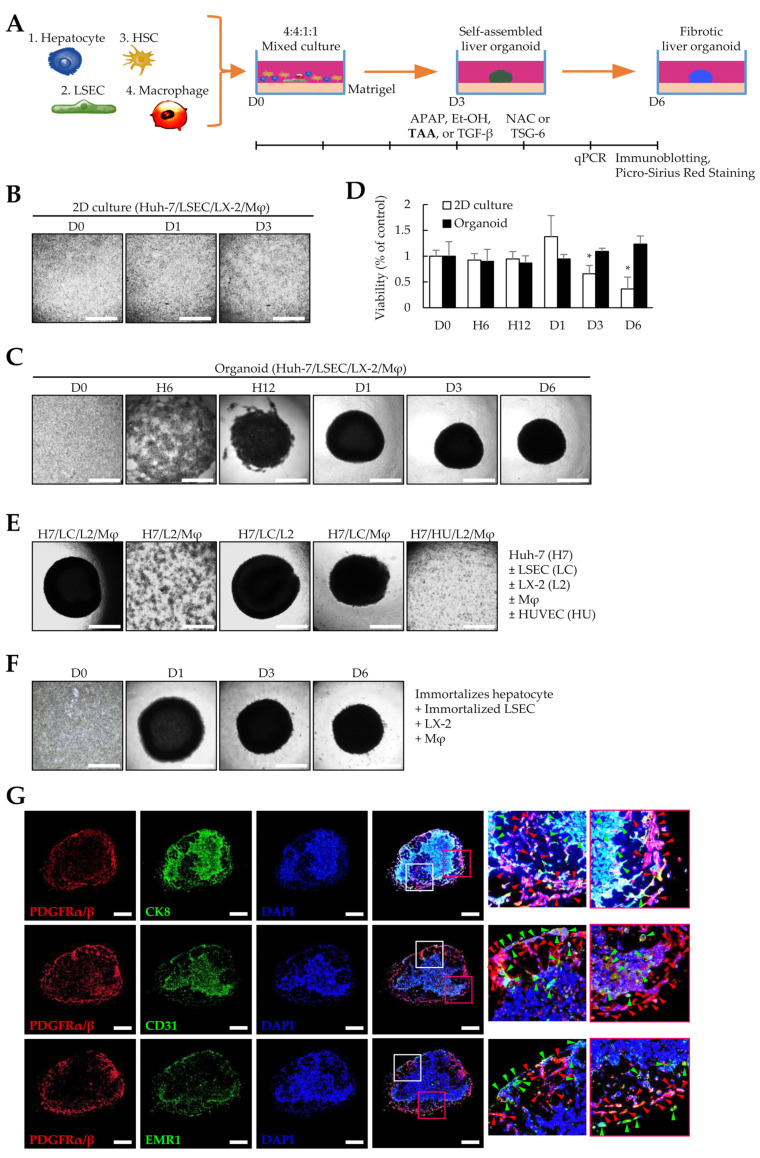
Liver organoids generated by multiple cells. To generate self-assembled liver organoids, Huh-7, liver sinusoidal endothelial (LSECs), LX-2, and macrophage cells were seeded in a Matrigel-coated 96-well plate and cultured for three days. Fibrotic liver organoids were produced by thioacetamide (TAA) treatment for an additional three days. (**A**) Schematic diagram of the experimental protocol. (**B**) Representative images of 2D-cultured cells at the indicated time points (days 0–3). The medium was replaced with fresh medium every two days. (**C**) Representative images at indicated time points (days 0 to 6). The self-assembled liver organoid was cultured for six days without subculturing while exchanging with fresh medium every two days. (**D**) Viability of 2D-cultured cells and organoids up to six days. * *p* < 0.05. (**E**) Cells required for self-assembly of liver organoids. Liver organoids were prepared by excluding one cell type from the Huh-7, LSECs, LX-2, and macrophage cells or by replacing LSECs with human umbilical vein endothelial cells (HUVECs). (**F**) Images of representative liver organoids generated using immortalized primary hepatocytes and LSECs instead of Huh-7 and LSECs. Scale bar for (**B**,**C**,**E**,**F**): 1 mm. (**G**) Immunofluorescence images of liver organoids. To evaluate the cell distribution, antibodies against CK8, CD31, EMR1, and PDGFRα/β were applied to liver organoid sections obtained from day 6 without TAA treatment to detect Huh-7, LSECs, macrophages, and LX-2 cells, respectively. Red arrowheads indicate PDGFRα/β-positive cells. Green arrowheads indicate CK8-, CD31-, or EMR1-positive cells. Scale bars: 500 µm. Data shown for (**B**–**G**) are representative of three independent experiments.

**Figure 2 cells-12-02514-f002:**
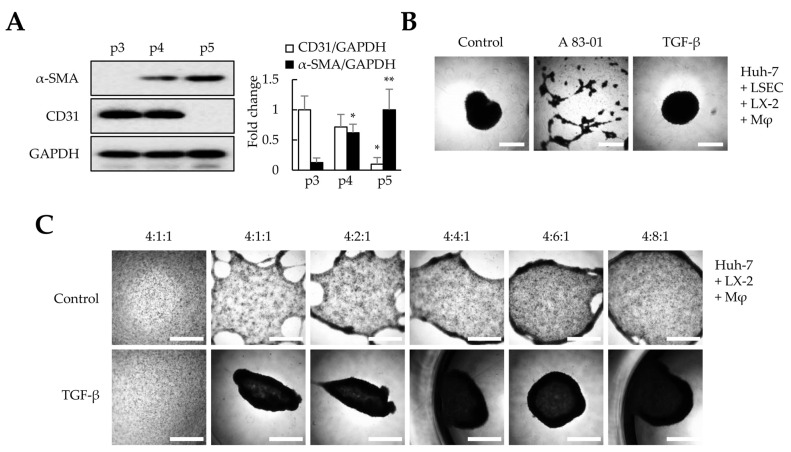
Effects of TGF-β signaling on liver organoid generation. In this study, we used liver sinusoidal endothelial cells (LSECs) at p4 and checked whether the LSECs at p4 had unique characteristics or were differentiated into fibroblasts; these aspects were evaluated by determining CD31 and α-SMA expression, respectively. In addition, to evaluate the effects of TGF-β signaling in LX-2 cells on liver organoid generation, multiple cells in Matrigel-coated plates were treated with A83-01 or TGF-β. Furthermore, we investigated whether LX-2 cells and TGF-β signaling affected liver organoid generation. (**A**) α-SMA and CD31 expression in LSECs according to the passage (p) number. The intensity of protein expression was quantified through densitometry in ImageJ, and its relative expression was normalized against that of GAPDH. * *p* < 0.05, ** *p* < 0.01. (**B**) Self-assembly of the four cell types (Huh-7, LSECs, macrophages, and LX-2 cells). A83-01, a TGF-β receptor inhibitor, inhibited the self-assembly of the four cell types. (**C**) Self-assembly of three cell types (Huh-7, macrophages, and LX-2 cells). In the absence of LSECs, a higher ratio of LX-2 cells increased self-assembly but did not enhance the formation of spherical clusters. However, when the mixing ratio of LX-2 cells was 6-fold higher or more and when TGF-β was present, spherical clusters similar to liver organoids formed when the four cell types were induced. Scale bars: 1 mm. Data shown are representative of three independent experiments.

**Figure 3 cells-12-02514-f003:**
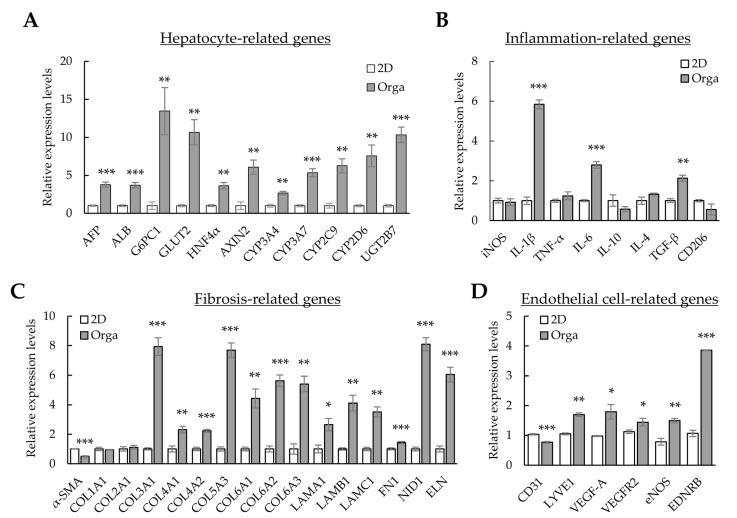
Gene expression profiles in 2D- and 3D-cultured cells. The four cell types were cultured for three days in 96-well plates without or with Matrigel for 2D or 3D cultures, respectively. Gene expression was analyzed using qPCR. (**A**) Hepatocyte-related gene expression profiles in 2D- and 3D-cultured cells. (**B**) Inflammation-related gene expression in 2D- and 3D-cultured cells. (**C**) Extracellular-matrix-related gene expression in 2D- and 3D-cultured cells. (**D**) Endothelial-cell-related gene expression in 2D- and 3D-cultured cells. Target gene expression was normalized using *GAPDH* expression. Relative fold changes in mRNA expression were measured using the 2^−(ΔΔCt)^ method. The results are presented as the means ± standard deviation (SD) of three replicates. * *p* < 0.05, ** *p* < 0.01, and *** *p* < 0.001. 2D; two-dimensional-cultured cells, Orga; 3D-cultured liver organoid.

**Figure 4 cells-12-02514-f004:**
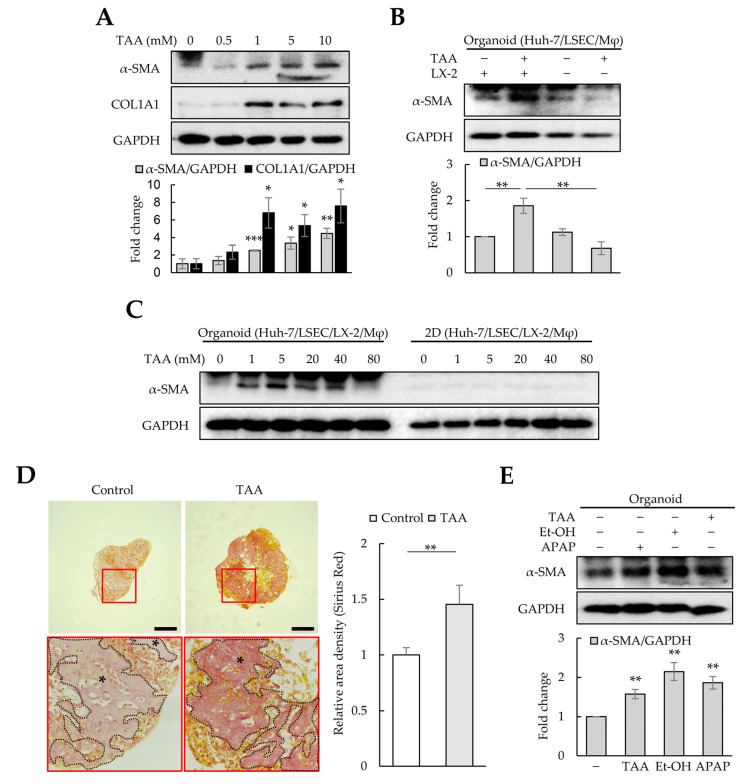
Fibrosis modeling in liver organoids using liver toxicants. Liver organoids on day 3 were treated with thioacetamide (TAA), ethanol (Et-OH), or acetaminophen (APAP) for an additional three days to generate fibrotic liver organoids. (**A**) α-SMA and COL1A1 expression in TAA-treated liver organoids. The intensity of protein expression was quantified through densitometry in ImageJ, and its relative expression was normalized against that of GAPDH. (**B**) LX-2-dependent α-SMA expression in fibrotic liver organoids. In liver organoids prepared without LX-2 cells, no increase in α-SMA expression was detected even after TAA treatment. (**C**) Liver-organoid-dependent α-SMA expression after TAA treatment. α-SMA expression increased in liver organoids treated with ≤5 mM TAA, but gradually decreased at >20 mM TAA, reaching control levels at 80 mM TAA. However, no α-SMA expression was detected in 2D-cultured cells treated with TAA up to 80 mM. Data shown are representative of two independent experiments. (**D**) Collagen expression in liver organoids. Liver organoid sections obtained on day 6 were stained with a Picro-Sirius Red solution to detect collagen expression with or without TAA treatment, and images were analyzed using light microscopy. The black stars indicate areas rich in collagen. Data shown are representative of three independent experiments. Scale bars: 50 µm. (**E**) α-SMA and COL1A1 expression in liver organoids treated with TAA, Et-OH, or APAP. Data in (**A**,**B**,**E**) are presented as the means ± SD of three independent experiments. * *p* < 0.05, ** *p* < 0.01, and *** *p* < 0.001.

**Figure 5 cells-12-02514-f005:**
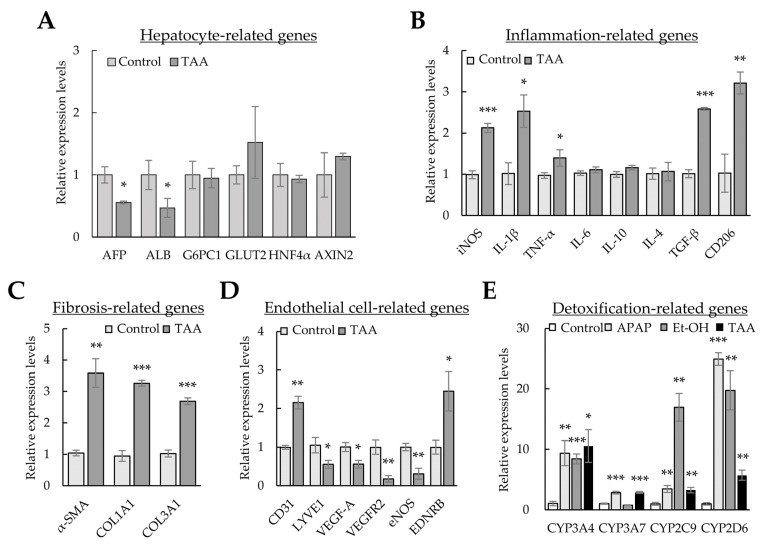
Gene expression profiles using liver toxicants. Liver organoids on day 3 were treated with the liver toxicants thioacetamide (TAA), ethanol (Et-OH), and acetaminophen (APAP) for an additional two days. Gene expression was analyzed using qPCR. (**A**) Hepatocyte-related gene expression profiles in TAA-treated liver organoids. (**B**) Inflammation-related gene expression in TAA-treated liver organoids. (**C**) Extracellular matrix (ECM)-related gene expression in TAA-treated liver organoids. (**D**) Endothelial-cell-related gene expression in TAA-treated liver organoids. (**E**) Expression of *CYPs* in toxicant-treated liver organoids. *GAPDH* expression was used to normalize the expression of target genes. The 2^−(ΔΔCt)^ method measured relative fold changes in mRNA expression. The results are presented as the means ± SD from three replicates. * *p* < 0.05, ** *p* < 0.01, and *** *p* < 0.001.

**Figure 6 cells-12-02514-f006:**
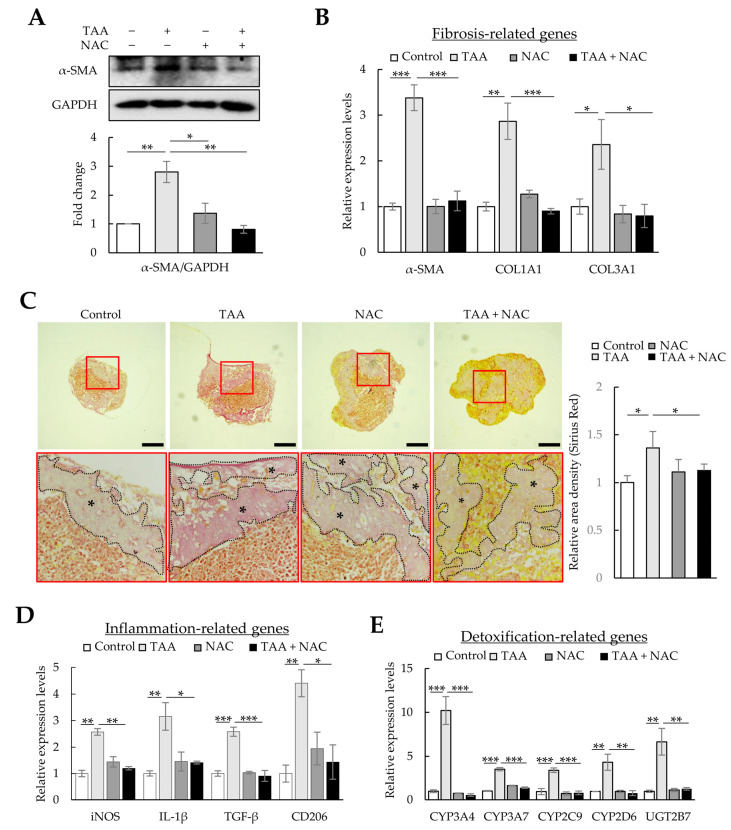
Antifibrotic effects of N-acetyl-L-cysteine (NAC) in thioacetamide (TAA)-treated liver organoids. Liver organoids on day 3 were treated with 1 mM of TAA, and NAC was added on day 4. qPCR, immunoblotting, and Picro-Sirius Red staining were conducted on samples prepared on day 5 or 6. (**A**) α-SMA and COL1A1 expression in fibrotic liver organoids treated with NAC. The intensity of protein expression was quantified through densitometry in ImageJ, and its relative expression was normalized against that of GAPDH. Data are presented as the means ± SD of three independent experiments. * *p* < 0.05 and ** *p* < 0.01. (**B**) Fibrotic marker expression in fibrotic liver organoids treated with NAC. The expression of *α-SMA*, *COL1A1*, and *COL3A1* was analyzed using qPCR on day 5. (**C**) Collagen expression in fibrotic liver organoids treated with NAC on day 6. Collagen in liver organoids was stained with Picro-Sirius Red solution, and images were analyzed using light microscopy. Data shown are representative of three independent experiments. The black stars indicate areas rich in collagen. Scale bars: 50 µm. (**D**) Inflammation-related gene expression in fibrotic liver organoids treated with NAC. (**E**) *CYP* expression in fibrotic liver organoids treated with NAC. qPCR (**B**,**D**,**E**) was conducted on day 5 samples, and target gene expression was normalized using *GAPDH* expression. Relative fold changes in mRNA expression were measured using the 2^−(ΔΔCt)^ method. The results are presented as the means ± SD of three replicates. * *p* < 0.05, ** *p* < 0.01, and *** *p* < 0.001.

**Figure 7 cells-12-02514-f007:**
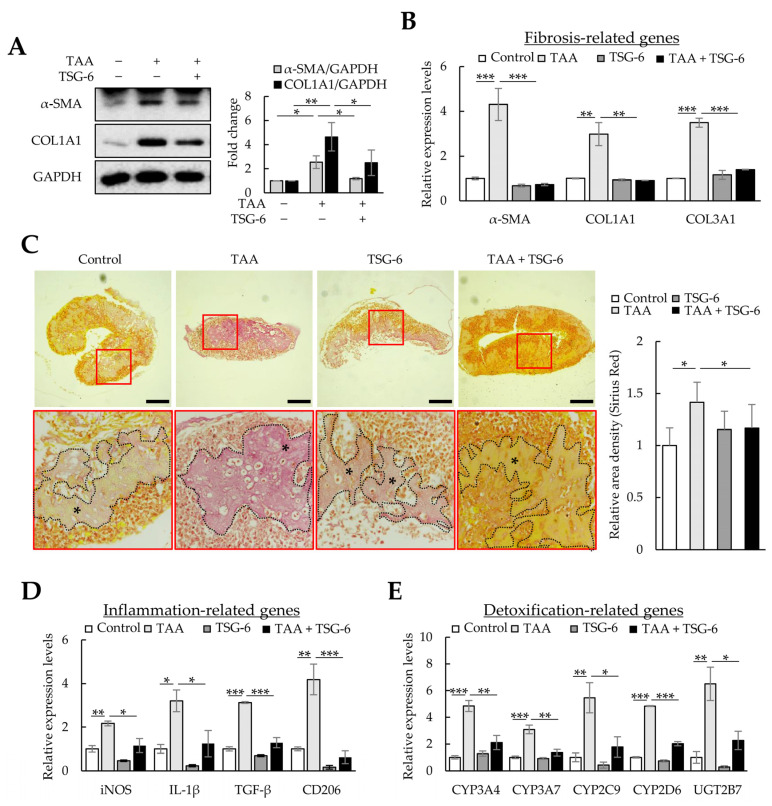
Antifibrotic effects of tumor necrosis factor-stimulated gene 6 protein (TSG-6) in thioacetamide (TAA)-treated liver organoids. Liver organoids on day 3 were treated with 1 mM of TAA, and TSG-6 was added on day 4. qPCR or immunoblotting and Picro-Sirius Red staining were conducted on samples prepared on day 5 or 6, respectively. (**A**) α-SMA and COL1A1 expression in fibrotic liver organoids treated with TSG-6. Data are presented as the means ± SD of three independent experiments. * *p* < 0.05 and ** *p* < 0.01. (**B**) Fibrotic marker expression in fibrotic liver organoids treated with TSG-6. The expression of *α-SMA*, *COL1A1*, and *COL3A1* was analyzed using qPCR on day 5. (**C**) Collagen expression in fibrotic liver organoids treated with TSG-6 on day 6. Collagen in liver organoids was stained with Picro-Sirius Red solution, and images were analyzed using light microscopy. The black stars indicate areas rich in collagen. Data shown are representative of three independent experiments. Scale bars: 50 µm. (**D**) Inflammation-related gene expression in fibrotic liver organoids treated with TSG-6. (**E**) *CYP* expression in fibrotic liver organoids treated with TSG-6. qPCR (**B**,**D**,**E**) was conducted on day 5 samples, and target gene expression was normalized using *GAPDH* expression. Relative fold changes in mRNA expression were measured using the 2^−(ΔΔCt)^ method. The results are presented as means ± SD of three replicates. * *p* < 0.05, ** *p* < 0.01, and *** *p* < 0.001.

**Figure 8 cells-12-02514-f008:**
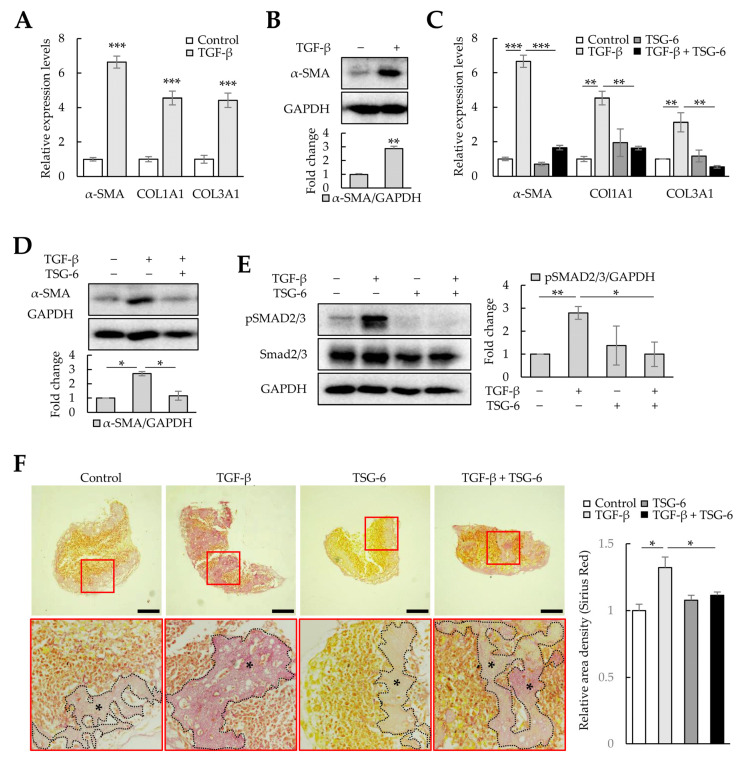
Antifibrotic effects of tumor-necrosis-factor-stimulated gene 6 protein (TSG-6) in TGF-β-treated liver organoids. Liver organoids on day 3 were treated with TGF-β, and TSG-6 was added on day 4. qPCR or immunoblotting and Picro-Sirius Red staining were conducted on samples prepared on day 5 or 6, respectively. (**A**) Fibrotic marker expression in TGF-β-treated liver organoids. The expression of *α-SMA*, *COL1A1*, and *COL3A1* was analyzed using qPCR on day 5. (**B**) α-SMA expression in TGF-β-treated liver organoids. (**C**) Antifibrotic effects of TSG-6 in TGF-β-treated liver organoids. The expression of *α-SMA*, *COL1A1*, and *COL3A1* was analyzed using qPCR. qPCR (**A**,**C**) was conducted on day 5 samples, and target gene expression was normalized using *GAPDH* expression. Relative fold changes in mRNA expression were measured using the 2^−(ΔΔCt)^ method. (**D**) Decreased expression of α-SMA by TSG-6 in TGF-β-treated liver organoids. (**E**) SMAD3 phosphorylation reduction by TSG-6 in TGF-β-treated liver organoids. Data in (**A**–**E**) are presented as the means ± SD of three independent experiments. * *p* < 0.05, ** *p* < 0.01, and *** *p* < 0.001. (**F**) Collagen expression in fibrotic liver organoids treated with TSG-6 on day 6. Collagen in liver organoids was stained with Picro-Sirius Red solution, and images were analyzed using light microscopy. Data shown are representative of three independent experiments. The black stars indicate areas rich in collagen. Scale bars: 50 µm.

## Data Availability

The data that support the findings of this study are available from the corresponding author upon reasonable request.

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
