# Peer review of "Generation of Fibrotic Liver Organoids Using Hepatocytes, Primary Liver Sinusoidal Endothelial Cells, Hepatic Stellate Cells, and Macrophages"

_cells, 2023, doi:10.3390/cells12212514_

Round 1
Reviewer 1 Report (New Reviewer)
I read the manuscript carefully: it is a high-quality, well-written work, relevant for the field and presented in a well-structured manner.
The manuscript cover a topic that will surely gain in importance in the coming years.
What is certainly interesting is the deep knowledge in the methods and techniques of organoid construction: the methods were explained thoroughly and cleanly
Very interesting the use of immunofluorescence to assess the distribution of various cellular subgroups to better understand organoid organization, and gene expression profile to evaluate the mechanism of fibrosis and detoxification could be similar to in liver organoid
As written in the discussion , this work lays the groundwork to create well-applicable models for research and study of pathophysiology of liver diseases.
Just a note: in the introduction, "As mesenchymal stem cells (MSCs) reduce inflammation," I would try to better explain the concept, which is true anyway.
Author Response
Please see the attachment.

Reviewer 2 Report (New Reviewer)
The English text is well written, clear to understand and spelling and grammar looks fine. A minor editorial check of the text may be sufficient.
The authors performed a study investigating several conditions and cellular responses to drug/toxin exposure (TAA, APAP, Et-OH) and immunological modulation (TSG-6). The overall impression of the experimental part is good, but some points need to be clarified.
The approach to use liver organoids to investigate liver fibrosis is not new. The innovative part this study appears to be the use of a non-primary hepatic cell line (Huh-7) for organoid formation. The authors utilized primary endothelial cells and therefore the system cannot be claimed to be completely based on non-primary cells. However, this was not a claim raised by the authors in the manuscript but it may be relevant for defining the position of this research within existing knowledge.
One very related publication in this context is "Novel human hepatic organoid model enables testing of drug-induced liver fibrosis in vitro", by S. Leite e. al., Biomaterials 78 (2016), that is surprisingly not cited by the authors. One point that immediately catches the eye when comparing the two liver-organoid models is the short time of documented organoid stability that is only 7 days in the present manuscript and 21 days in the manuscript by Leite et al. The 7-day stability is in this paper even considered insufficient and motivated the authors to modify the co-culture to finally achieve a 21-day stability. Please comment on these points and add the according information also to the manuscript. I would like to add that the reviewer is not author or co-author of the paper by Leite and also not affiliated otherwise.
Repeated protein and gene expression experiments are shown as mean of three independent experiments or replicates (Fig.4, Fig.5). But how many different organoids have been analyzed for each condition? This is be important, because we cannot assume the relation of cell numbers to be totally consistent in each organoid.
The protein expression shown in Fig2A is surprising, as CD31 vanishes more or less completely in p5, while it was prominently expressed in p4. This could be explained by a relative long time between passages, where almost all cells changed their differentiation state between two passages. How long were the cells cultivated between passages? How does these time relate to spheroid formation time?
There is a strange looking smear on the western blot of alpha-SMA in Fig.4 A. Please show a different blot if available.
The abstract should be partly re-written: In line 19 the repeated occurrence of the word "using" should be avoided. Section between line 22-24 is strange because specific cell type related gene expression is reported without mentioning the used cell types before.
In line 39-41, all reported facts should be supported by citations. "Mesenchymal stem cells (MSCs) reduce inflammation" appears to be to a statement that is too general when out of context.
Round 2
Reviewer 2 Report (New Reviewer)
The authors responded to all points raised in the review and sufficiently improved the manuscript. I apologize for not seeing the citation of the work by Leite et al. that has been already been provided in the original manuscript as correctly stated by the authors.
The fact that the stability of the organoids is lasting for a period of 21 days or longer is very promising and I recommend at least to mention this fact in the text.
This manuscript is a resubmission of an earlier submission. The following is a list of the peer review reports and author responses from that submission.
Round 1
Reviewer 1 Report
In the present manuscript, Yoon and colleagues investigated the potentiality of liver organoids to model liver fibrosis. Organoids were assembled in matrigel by seeding hepatocytes, LSECs, hepatic stellate cells and macrophages. Then, organoid growth and phenotype was assessed, the latter by means of histology/immunofluorescence, western blot and RTqPCR.
The development of in vitro tools to model liver function is a relevant topic, especially for regenerative medicine and research purposes. However, the evidences provided by the Authors are not convincing enough regarding the validity of their culture method and soundness of results. Major points are reported below:
• Most images are at very low magnification, making it hard to appreciate the cells. Particularly, this applies to bright field images and immunofluorescence stains.
• Regarding IF, higher magnification is required in Figure 1E to appreciate cell morphology and actual marker expression. Moreover, double immunofluorescences are needed to show cell arrangements within the organoids.
• The Authors performed Sirius red stains to assess fibrosis and collagen deposition in TAA-treated organoids. However, the stains are not convincing. Images show large areas of amorphous tissue within the organoids, which are quite similar in width (at least apparently) among the different experimental groups, except for the intensity of SR stain. However, fibrosis would be characterized by the presence of definite fibers staining positive to sirius red, which are not present in these organoids. Moreover, no actual data or methods on SR quantification is provided.
• By sirius red stains, it appears that most cells within the organoids have lost their normal morphology. Particularly, hepatocytes are not apparent, and most cells seems damaged or altered in some way. Authors should address the methods of organoid processing and fixation, probably, in order to preserve (or confirm the presence of) normal cell morphology.
• Authors should include representative images and / or (when applicable) actual data from 2D cultures as well (e.g. some comparisons are reported, without actual supporting data).
• Since the macrophages in their culture condition are derived from differentiated monocytes, Authors should refrain from referring to them as "Kupffer cells" in the manuscript.
• The quality of western blot images is not very high and should be improved.
• Based on provided images, the size of the organoids increases dramatically at day 1 already, without apparent changes in the following days, which is a bit surprising. Authors should include actual data on organoid size, growth rate, and cell vitality/apoptosis/necrosis etc...
Reviewer 2 Report
The manuscript is clear and exciting results are presented. Nevertheless, minor changes are suggested.
1. In line 122, please indicate the concentration, fixed or range, used for TAA and TGF-Beta.
2. Lines 125 and 139. Were the organoids used for imaging and RNA isolation pooled from several wells, or was one well enough for each sample?
3. Line 126. Please indicate sonication conditions.
4. In line 145 indicate what gene was used as a reference.
5. NAC and TDG-6 treatment indicated in Figure 1A, are not mentioned in the methodology section, please include them.
6. Line 224. A83-01 treatment is not mentioned in the methods section, please include it.
7. Line 297 and Figure 4E) EtOH and APAP are not mentioned in the methods sections. Please include the details.
8. Ideally, change the color of the stars used on Picro-Sirius red staining images to one with more contrast, so is easier to see what is being indicated.